# Reproducibility and Predictive Value of White-Coat Hypertension in Young to Middle-Age Subjects

**DOI:** 10.3390/diagnostics13030434

**Published:** 2023-01-25

**Authors:** Paolo Palatini, Lucio Mos, Francesca Saladini, Olga Vriz, Claudio Fania, Andrea Ermolao, Francesca Battista, Marcello Rattazzi

**Affiliations:** 1Department of Medicine, University of Padova, 35128 Padova, Italy; 2San Antonio Hospital, 33038 San Daniele del Friuli, Italy; 3Cittadella Town Hospital, 35013 Cittadella, Italy; 4Villa Maria Hospital, 35138 Padova, Italy

**Keywords:** white-coat, hypertension, reproducibility, agreement, prognosis, young

## Abstract

(1) Aim. The aim of the study was to investigate the reproducibility of white-coat hypertension (WCH) and its predictive capacity for hypertension needing antihypertensive treatment (HT) in young to middle-age subjects. (2) Methods. We investigated 1096 subjects from the HARVEST. Office and 24 h blood pressures (BP) were measured at baseline and after 3 months. The reproducibility of WCH was evaluated with kappa statistics. The predictive capacity of WCH was tested in multivariate Cox models (N = 1050). (3) Results. Baseline WCH was confirmed at 3-month assessment in 33.3% of participants. Reproducibility was fair (0.27, 95%CI 0.20–0.37) for WCH, poor (0.14, 95%CI 0.09–0.19) for office hypertension, and moderate (0.47, 95%CI 0.41–0.53) for ambulatory hypertension. WCH assessed either at baseline or after 3 months (unstable WCH) was not a significant predictor of HT during 17.4 years of follow-up. However, participants who had WCH both at baseline and after 3 months (stable WCH) had an increased risk of HT compared to the normotensives (Hazard ratio, 1.50, 95%CI 1.06–2.1). (4) Conclusions. These results show that WCH has limited reproducibility. WCH diagnosed with two BP assessments but not with one showed an increased risk of future HT. Our data indicate that WCH should be identified with two sets of office and ambulatory BP measurements.

## 1. Introduction

A large body of evidence suggests that ambulatory blood pressure (BP) monitoring (ABPM) is superior to office BP for the management of hypertension. One of the main advantages of ABPM is the identification of patients with white-coat hypertension (WCH) whose BP is high in the doctor’s office but normal outside the medical environment [1,2,3,4,5]. Subjects with WCH have been found to be at lower risk than patients with sustained hypertension [6,7], and according to the opinion of some experts, they do not need antihypertensive treatment, especially if they are normotensive over the whole 24 h period [1,2,7,8]. However, the clinical importance of WCH remains controversial, because a number of studies have shown that patients with WCH may also be at increased risk of cardiovascular complications [4,5,9,10]. Part of the controversy may be due to the different methods used to identify people with WCH and to the limited reproducibility of this condition. In the Pressioni Arteriose Monitorate E Loro Associazioni (PAMELA) study, the risk of cardiovascular and total mortalities was not increased in people with WCH [11]. However, in the WCH patients with persistently elevated office BP, the risk of fatal events was greater than in a group of normotensive control subjects. Unfortunately, in that study, only office BP was remeasured after the first assessment, so the reproducibility of WCH could not be tested. In the European Lacidipine Study of Atherosclerosis (ELSA), only about 40% of treated hypertensive patients classified as white-coat uncontrolled hypertensive (WUCH) maintained the same classification at a subsequent evaluation [12]. In a group of treated hypertensive subjects with resistant hypertension, WCH diagnosis presented a moderate reproducibility and was confirmed in 144 out of 198 patients after 3 months [13]. A fair reproducibility of WCH was found by Ben-Dov et al. in a mixed population of treated and untreated patients attending an outpatient clinic, but in that study, only 31 WCH subjects were included [14]. Little information is available on the reproducibility of WCH in untreated subjects. In the Spanish Registry, WCH reproducibility was high when the period between the two assessments was <1 month, whereas the reproducibility was low when the assessments were separated by more than 1 month [15]. However, whether the prognostic capacity of confirmed WCH (stable WCH) was better than that of unstable WCH was not investigated.

The aim of the present study was twofold. Firstly, it was to assess the reproducibility of WCH measured twice three months apart in a cohort of untreated young to middle-age participants; and secondly, it was to verify whether stable WCH was a more accurate predictor of established hypertension than unstable WCH during a long-term follow-up.

## 2. Methods

The Hypertension and Ambulatory Recording Venetia Study (HARVEST) is a prospective cohort study, initiated in April 1990 in the northeast of Italy with the collaboration of 17 hypertension units [16,17,18]. To be enrolled, participants had to be 18 to 45 years old, have stage 1 hypertension at entry, and have never been treated for hypertension before. In all participants, the presence of diabetes, renal impairment, cardiovascular abnormalities, or other serious diseases was excluded at the baseline [16,17,18]. For the present analysis, we examined 1096 participants who had performed two office and ambulatory BP assessments within 3 months after enrolment. Of these, 1050 subjects who had long-term follow-up data were considered for the survival analysis.

### 2.1. Procedures

The procedures followed were in accordance with institutional guidelines. Baseline data included a medical and family history and a self-compiled questionnaire about tobacco use, alcohol and coffee consumption, and physical activity habits. Details about these lifestyle factors have been reported elsewhere [19]. Baseline brachial office BP was the mean of six measurements obtained with the auscultatory measurement with appropriately sized cuffs, during two visits performed two weeks apart. After the initial screening, participants also underwent 24 h ABPM, using the A&D TM2420 model 7 (A&D, Tokyo, Japan) or ICR Spacelabs 90207 monitor (Spacelabs, Redmond, WA) devices. Both of these devices were previously validated [20,21]. Measurements were taken every 10 min during the day (06.00–23.00 h) and every 30 min during the night (23.00–06.00 h) according to the previously published procedures [17]. Average 24 h BP was calculated as the mean of the individual means calculated for each hour. The study was approved by the HARVEST Ethics Committee and by the Institutional Review Board of the Department. A written informed consent was given by the participants.

### 2.2. Definition of Operational Threshold Level

In the HARVEST study, the primary end point is the development of hypertension needing antihypertensive treatment according to the guidelines criteria for young subjects at low cardiovascular risk available at the time of patient assessment. The operational threshold level for identifying participants who reached the end point therefore changed over time in keeping with available guidelines. In 1990, when we started the study, the British Hypertension Society stipulated that eligibility for antihypertensive medication was determined by progression to grade II hypertension (supine office systolic blood pressure ≥160 mm Hg and/or supine office diastolic blood pressure ≥100 mm Hg) during the first year of follow-up [22]. Later on, the 1999 ISH/WHO guidelines for patients at low cardiovascular risk, such as the participants in the present study, established that treatment should be given to subjects with a supine office systolic blood pressure ≥150 mm Hg and/or supine office diastolic blood pressure ≥95 mm Hg in two consecutive visits [23]. After the publication of the 2003 European Society of Hypertension–European Society of Cardiology guidelines for the management of arterial hypertension [24], which also adopted the 140/90 mmHg cut-off for subjects at low risk, we finally used the 140/90 mmHg threshold. The diagnosis of hypertension was thus based on repeated office BP measurement as recommended by the guidelines and not on ambulatory BP. Thus, people with masked hypertension were not treated.

### 2.3. Follow-Up

Follow-up visits were scheduled after 1, 2, 3, and 6 months and thereafter at 6-month intervals. If, after at least 6 months of implementation of non-pharmacological measures, the participant’s office BP was still above the “operational threshold level”, the patient was rescheduled for a visit within 2 to 4 weeks and the average office BP was calculated. If office BP was persistently above the operational threshold level, the patient was given antihypertensive drug treatment; otherwise, he or she was checked at monthly intervals. Then, treated and untreated subjects continued to be checked at 6-month intervals. For the present analysis, only untreated participants were considered. Other details on follow-up procedures in the HARVEST were reported elsewhere [16,17,18,19]. 

### 2.4. Patients’ Classification

At the baseline, all participants had elevated office BP, and thus, they could be categorized as WCH or sustained hypertensives, according to their office and 24 h BP levels. The cutoff values used to define normal and high BPs were 140/90 mmHg for office BP and 130/80 mmHg for 24 h BP, respectively. Due to spontaneous changes of office and 24 h BPs, 3 months after enrolment, we could identify four different groups: (1) people with normal office and 24 h BPs (normotensives); (2) people with high office BP and normal 24 h BP (WCH subjects); (3) people with normal office BP and high 24 h BP (masked hypertensives); (4) people with high office and high 24 h BP (sustained hypertensives). In addition, people with WCH were further subdivided according to whether WCH was found both at baseline and repeat assessment (stable WCH) or was present only at one of the two assessments (unstable WCH). White-coat effect (WCE) was defined as the difference between office BP and average 24 h BP.

### 2.5. Statistics

Quantitative variables were reported as mean and SD, or as median and interquartile range (IQR), and differences in the distribution across groups were tested by one-way ANCOVA adjusting for age and sex. Categorical variables were reported as percentage, and differences in the distribution were tested by χ^2^ test. For correlations, Pearson’s test was used with Bonferroni correction. The reproducibility of WCH, office hypertension, and ambulatory hypertension was evaluated using kappa statistics, which measure agreement occurring in excess of that expected by chance. Weighted Kappa (WK) was calculated according to Cohen’s method using linear weight [25]. The standard error and 95% confidence interval were calculated according to Fleiss et al. [26]. According to Altman, the strength of agreement can be defined as poor if kappa is <0.20, fair if kappa is 0.21–0.40, moderate if kappa is 0.41–0.60, good if kappa is 0.61–0.80, and very good if kappa is 0.81–1.00 [27]. The risk of development of hypertension requiring pharmacological treatment (defined herein as “established hypertension”) in relation to the BP category was evaluated by means of multivariable Cox analyses, adjusting for risk factors and confounders. Analyses were performed using Systat version 12 (SPSS Inc., Evanston, IL, USA) and MedCalc version 15.8 (MedCalc Software, Ostend, Belgium).

## 3. Results

Mean ± SD BP at entry was 145.8 ± 10.5/93.7 ± 5.8 mmHg, mean age was 33.0 ± 8.6 years, and mean BMI was 25.4 ± 3.5 kg/m^2^. Due to the natural selection of people with high BP in this particular age range, there was a higher prevalence of males (*n* = 797; 72.7%). At entry, 228 participants (20.8%) had WCH and 868 (79.2%) had sustained hypertension. The age- and sex-adjusted WCE was greater in the WCH participants than the rest of the population (mean ± SEM = 24.5 ± 0.8/19.2 ± 0.5 mmHg vs. 12.1 ± 0.4/10.6 ± 0.2 mmHg, *p* < 0.001/< 0.001).

### 3.1. Follow-Up BP Changes

After 3 months of follow-up, mean office BP in the whole group fell to 140.5 ± 12.1/90.5 ± 8.6 mmHg (*p* < 0.001/< 0.001 versus baseline). Average 24 h BP showed only a small non-significant decline from 131.1 ± 10.8/81.4 ± 8.3 mmHg to 130.6 ± 11.0/80.9 ± 8.4 mmHg (*p* = 0.33/= 0.20). Correlation coefficients between baseline and repeat office SBP/DBP were 0.46/0.45 (both *p* < 0.001) and between baseline and repeat 24 h SBP/DBP were 0.73/0.71 (both *p* < 0.001). At the 3-month assessment, the WCE declined in all subjects and was still greater in the WCH participants than the rest of the population (mean ± SEM = 15.3 ± 0.9/12.7 ± 0.6 mmHg vs. 8.5 ± 0.4/8.8 ± 0.3 mmHg, *p* < 0.001/< 0.001). However, the WCE decline from baseline was greater in the former than the latter (mean ± SEM = 9.3 ± 0.9/6.5 ± 0.6 mmHg vs. 3.6 ± 0.5/1.8 ± 0.3 mmHg, *p* < 0.001/< 0.001).

On the basis of office and average 24 h BPs, at the 3-month assessment, we could identify 132 subjects with both normal office and 24 h BPs, 159 subjects with WCH, 195 subjects with masked hypertension, and 610 subjects with sustained hypertension. Among the 159 subjects with WCH at repeat assessment, only 76 also had WCH at baseline (stable WCH). The characteristics of the study participants by BP category at the 3-month assessment are reported in Table 1. Apart from office and ambulatory BP levels, only small differences in clinical characteristics were present between the groups. In particular, lifestyle habits did not differ significantly across the BP categories. Follow-up duration was shorter in stable WCH than in unstable WCH.

Baseline WCH was confirmed at 3-month assessment in only 33.3% of participants (Figure 1), whereas 31.6% became normotensive, 8.8% masked hypertensive, and 26.3% sustained hypertensive. WCH reproducibility was slightly better for the women (38.4% at second assessment) than the men (31.0%), but the between-sex difference was not significant (*p* = 0.68).

Twenty-one point four percent of WCH patients whose office BP normalized after three months had high ambulatory BP at repeat ABPM. WCH reproducibility evaluated with weighted kappa was fair (WK, 0.27, 95%CI 0.20–0.37). Office hypertension (BP ≥ 140/90 mmHg) showed a poor agreement (WK, 0.14, 95%CI 0.09–0.19), and ambulatory hypertension (24 hBP ≥ 130/80 mmHg) showed a moderate agreement (WK, 0.47, 95%CI 0.41–0.53).

### 3.2. WCH as Predictor of Hypertension Needing Antihypertensive Treatment

Long-term follow-up data were available in 1050 participants. During 17.4 years of follow-up, 80.5% of participants developed established hypertension. In a multivariable Cox regression analysis including age, sex, BMI, smoking, alcohol and coffee use, and physical activity habits, unstable WCH assessed either at baseline (*p* = 0.86) or after 3 months (*p* = 0.16) was not a significant predictor of future established hypertension. In contrast, participants who had WCH both at baseline and after 3 months (N = 76) had an increased risk compared to the normotensives (hazard ratio 1.50, 95%CI 1.06–2.13, *p* = 0.022) (Figure 2). Hazard ratios were 1.35 (95%CI 1.02–1.80, *p* = 0.037) in the participants with masked hypertension and 1.52 (95%CI 1.19–1.95, *p* < 0.001) in those with sustained hypertension (Figure 2).

## 4. Discussion

The present study shows that WCH is not a stable condition in young to middle-age people, as in our cohort WCH could be confirmed only in one-third of the subjects at second assessment. Two-thirds of patients with WCH at baseline showed a different diagnosis at repeat assessment, which, in most cases, was normotension or sustained hypertension. According to Cohen’s kappa statistics, WCH reproducibility was fair (WK, 0.27), chiefly due to a poor agreement of office hypertension over the 3 months of observation. Only stable WCH was a significant predictor of future established hypertension, with a risk level similar to that conferred by masked or sustained hypertension.

### 4.1. WCH Reproducibility

The reproducibility of WCH has been evaluated mostly in patients with treated hypertension, a condition called white-coat uncontrolled hypertension (WUCH). In the ELSA study, only 38–45% of patients exhibiting WUCH at the first set of office and ambulatory BP measurements remained in the same condition one year later [12]. A better reproducibility was found in a group of patients with white-coat resistant hypertension, in which the diagnosis of WUCH was confirmed in 73% after 3 months, with the remaining patients developing sustained hypertension [13]. In a retrospective study of a mixed population of treated and untreated patients, with a mean interval between measurements of 1.5 years, only 14 of 31 patients (45%) with WCH or WUCH at the time of the first assessment had the same diagnosis at the time of the second assessment [14]. In treated patients, BP differences between measurements may obviously be influenced by changes in treatment. 

Only few reproducibility data are available in untreated individuals. In 839 untreated patients from the Spanish Registry, who underwent two separate office BP and ABPM assessments, the proportion of patients falling into the WCH category in the two assessments was 55% [15]. However, the reproducibility of this hypertension phenotype was highly dependent on the time elapsed between the assessments. Agreement was good when the between-measurement period was shorter than 1 month (89%), whereas the reproducibility was worse when assessments were separated by more than 1 month (42%). In the Spanish study, most subjects with unstable WCH developed sustained hypertension at repeat assessment (25.9%), in agreement with the present results (26.3%) [15]. However, in our study, almost one-third of participants with WCH at baseline were found to be normotensive after 3 months. This was due to the important decrease in office BP on going from the first to the second measurement and to the poor agreement of clinic hypertension over the two assessments. It is possible that non-pharmacologic therapies implemented soon after enrolment favoured changes in office BP during the first 3 months. However, the marginal decrease in 24 hBP after 3 months of observation argues against this interpretation. Our participants were seen every 1 month during the initial period, and a progressive attenuation of the white-coat reaction with repeated measurements is likely to account for the office BP reduction after 3 months. Although people with WCH also had a greater WCE than the rest of the population at second assessment, the WCE declined more in the former than the latter.

### 4.2. Predictive Value of WCH

The clinical significance of WCH is still controversial. Interpretation of longitudinal data from cohort studies on the predictive value of WCH for adverse events has been complicated by inclusion of treated subjects, lack of uniformity of diagnostic criteria, and differing characteristics of the participants [1,2,3,4,5]. According to some investigators, WCH should be classified as a ‘prehypertensive’ state, because some studies in middle-age and older subjects have shown that individuals with WCH have increased risk of developing sustained hypertension when compared with truly normotensive subjects [28,29]. However, in those studies, the predictive value of stable versus unstable WCH was not tested. In the subjects of the PAMELA population in whom WCH was diagnosed with a single set of office BP measurements, the risk of cardiovascular and all-cause mortality was not significantly different from that of the normotensive subjects of control [11]. However, in the participants in whom an elevation of office BP was found over two visits performed before and after the ABPM, the risk of both cardiovascular and all-cause mortality was significantly increased. In the present study of young subjects screened for stage 1 hypertension exhibiting a marked white-coat effect at enrolment, a single detection of WCH showed low predictive capacity for the development of established hypertension in future years. However, WCH did predict the development of established hypertension when baseline and 3-month diagnoses were concordant. Our data are in agreement with those obtained by Miyashita et al. in a pediatric population of 89 subjects (median age, 13.9 years; 78% male) in whom two ABPM measurements were performed 14 months apart [30]. Individuals with stable WCH were more likely to progress to either prehypertension or ambulatory hypertension than those with unstable WCH. Overall, these data indicate that two sets of BP measurements should be performed for a correct identification of the WCH phenotype.

### 4.3. Limitations

Several limitations of this study should be acknowledged. First, our participants were not selected from a general population but from a population of subjects who were referred to hypertension units for stage 1 hypertension and whose classification was based on office BP and ABPM performed after 3 months of observation. In addition, the present results were obtained in people 45 years of age or younger and are not generalizable to older people. A further limitation may be the length of time elapsed between the two assessments. Changes in lifestyle after the baseline visit may have influenced BP during the 3-month period. On the other hand, from a clinical standpoint, this time interval appears to be a reasonable compromise. Previous analyses of the HARVEST showed that only minor changes in participants’ lifestyle habits occurred during the long-term follow-up [31]. In addition, in the long-term analysis, the definition of our end point changed over time, because the criteria for treating hypertension varied from 1990 to 2003. The follow-up length was shorter in the stable than in the unstable WCH. However, this might have reduced rather than increased the difference in risk compared to the normotensive or the unstable WCH group. Finally, we report data only from Caucasians, which may not be applicable to other ethnic groups.

## 5. Conclusions

The present data show that in many subjects screened for stage 1 hypertension, there is a rapid decline in office BP during the first few months of observation due to a pronounced white-coat effect at baseline examination, which tends to attenuate over time. Thus, in a subset of these individuals, WCH might be just an occasional BP pattern. In this BP phenotype, it is thus mandatory to measure office BP repeatedly before therapeutic decisions are made. On the other hand, 21.4% of our WCH subjects whose office BP normalized had elevated ambulatory BP after 3 months, suggesting that ABPM should also be repeated to better define BP status. Our data also suggest that it will be important to improve the methodology of BP assessment in longitudinal observational studies, which have almost always been based on a single set of office and ambulatory BP measurements.

## Figures and Tables

**Figure 1 diagnostics-13-00434-f001:**
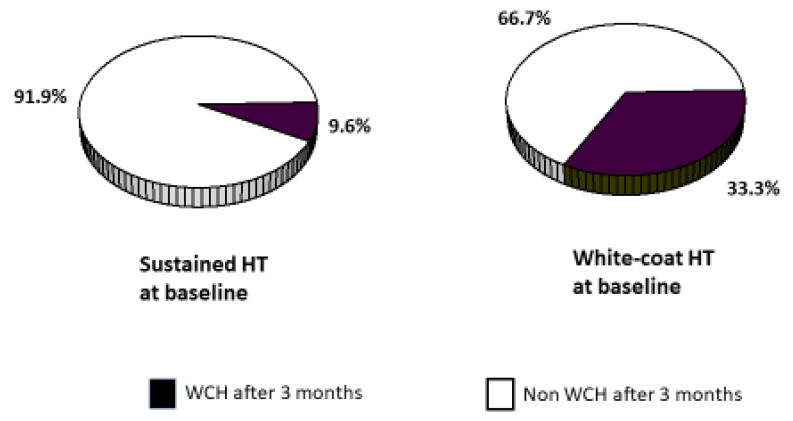
Frequency of white-coat hypertension at second blood pressure assessment performed three months after enrolment, in the participants stratified according to whether they had WCH or sustained hypertension at baseline. HT indicates hypertension; WCH indicates white-coat hypertension.

**Figure 2 diagnostics-13-00434-f002:**
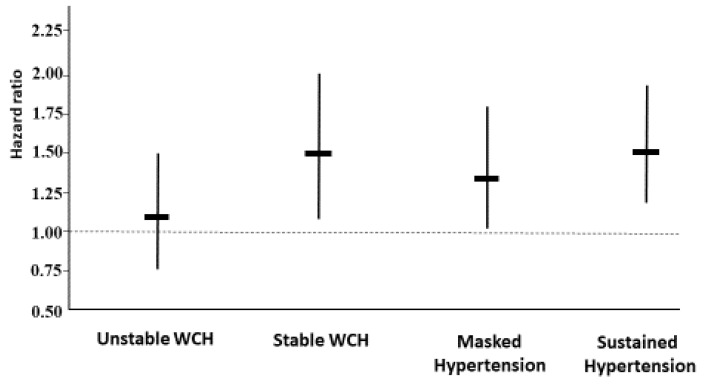
Long-term risk of established hypertension in the participants stratified according to their BP status assessed three months after enrolment. Hazard ratios and 95% confidence intervals were derived from a multivariable Cox regression model. The normotensive group was taken as the reference. WCH indicates white-coat hypertension.

**Table 1 diagnostics-13-00434-t001:** Clinical characteristics of the 1096 participants by BP status assessed three months after enrolment.

	Normotensive (*n* = 132)		Masked HT (*n* = 195)		Sustained HT (*n* = 610)		Stable WCH (*n* = 76)		Unstable WCH (*n* = 83)		
Variable	Mean	SD	Mean	SD	Mean	SD	Mean	SD	Mean	SD	*p*-value
Age, years	32.1	8.3	32.9	8.3	33.5	8.8	31.7	7.7	32.3	8.2	0.21
BMI, kg/m^2^	24.4	2.9	24.8	3.2	25.8	3.4	25.4	4.6	25.3	3.4	0.002
Office SBP, mmHg	128.0	7.6	129.6	7.6	145.8	9.9	143.9	11.8	144.2	9.8	<0.001
Office DBP, mmHg	81.5	6.7	83.6	6.0	94.0	7.0	93.8	7.9	92.3	6.7	<0.001
24 h-SBP, mmHg	118.9	7.2	132.9	8.6	135.0	9.7	118.7	6.6	122.1	5.8	<0.001
24 h-DBP, mmHg	73.3	5.5	81.7	7.7	84.0	7.6	74.2	5.4	74.3	6.0	<0.001
24 h-HR, bpm	70.9	7.4	72.5	7.1	72.6	7.7	72.8	8.1	71.0	9.2	0.009
FU length, years *	18.2	12.0–23.3	16.0	7.8–23.6	17.4	9.1–22.3	15.7	8.1–20.4	19.0	11.9–25.1	0.047
Sex, men	62.9%	--	71.8%	--	75.9%	--	63.2%	--	75.9%	--	0.009
Smoking, yes	16.7%	--	21.0%	--	22.8%	--	20.6%	--	15.7%	--	0.20
Alcohol use, yes	37.9%	--	46.7%	--	50.2%	--	39.5%	--	48.2%	--	0.07
Coffee use, yes	70.5%	--	66.7%	--	76.2%	--	71.1%	--	75.9%	--	0.09
Physical activity, yes	40.9%	--	41.0%	--	36.1%	--	38.2%	--	47.0%	--	0.30

Data are mean and standard deviation or percent. HT indicates hypertension; BMI, body mass index; WCH, white-coat hypertension; SBP, systolic blood pressure; DBP, diastolic blood pressure; HR, heart rate; 24 h, average over 24 h; FU, follow-up. * Median and IQR.

## Data Availability

The data that support the findings of this study are available on reasonable request from the corresponding author.

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
