# Peer review of "Reproducibility and Predictive Value of White-Coat Hypertension in Young to Middle-Age Subjects"

_diagnostics, 2023, doi:10.3390/diagnostics13030434_

Round 1
Reviewer 1 Report
Palatini and coworkers analyzed in a large cohort the reproducibility of white-coat hypertension (WCH) and its predictive capacity for hypertension needing antihypertensive treatment in young-to-middle-age subjects. They found that white-coat hypertension has limited reproducibility and only if confirmed at a subsequent ambulatory blood pressure monitoring (ABPM) has a predictive value for the risk of developing sustained hypertension in the future.
The study has considerable relevance in putting under the correct spotlight the scarce reproducibility of white-coat hypertension diagnosis. However, the authors should also consider and resolve some relevant methodological issues:
1. It is not clear the criteria for selecting patients: the authors state that patients underwent calculation of average BP (it is supposed by ABPM) if the participant’s BP was still above the “operational threshold level” (that means with repeated findings of elevated office blood pressure). However, after a few paragraphs, they report a group of normotensive subjects, described as “people with normal office and 24-hour BP”. Please explain this incoherence.
2. The authors state that “for the present analysis only untreated participants were considered”. Does this mean that subjects with sustained hypertension at first ABPM were not treated until confirmation of this condition at the three-months follow-up? Please explain or better detail the used protocol.
3. The authors should also consider the potential role of a time-dependent selection bias: patients at higher risk (with the higher operational threshold to access BPM, according to the 1990 guidelines) have the longer follow-up and might disproportionately contribute to the development of sustained hypertension at follow-up. Authors should specify the composition of the study cohort according to enrolment criteria and perform a sensitivity analysis restricting the analyses to patients managed with homogeneous criteria (i.e. after 2003)
4. Figure 2 does not represent data as conventionally done: since data represent complementary percentages, the correct representation should be as pie charts and not as bars.
5. “Established hypertension” at follow-up was defined as “hypertension requiring pharmacological treatment”. This kind of approach is highly prone to the subjective perception of the physician regarding the need for treatment and is poorly standardizable. Therefore, this endpoint is questionable and should be substituted by a more objective criterion (e.g. blood pressure level). Moreover, this endpoint does not necessarily represent an index of the severity of the disease. Authors should better direct the survival analyses to the search for the occurrence of more clinically relevant events (i.e. MACES).
Reviewer 2 Report
I found the present research quite interesting and I just have few consideration
Since gender is an effective factor in blood pressure how did you adjust this factor?
Did you have any exclusion criteria?
best regards
Reviewer 3 Report
This article by Palatini et al investigates the reproducibility of white-coat hypertension and its predictive capacity for hypertension needing antihypertensive treatment. The study is clearly written and interesting. One minor concern: did lifestyle changes occur in the long term followup period? If so, were they taken into consideration? Because I have noticed that one limitation is that “Changes in lifestyle after the baseline visit may have influenced BP during the 3-month period”. What about the longterm- 17.4 years?
Round 2
Reviewer 1 Report
I would thank the authors for their reply and for their efforts to improve the manuscript by addressing all the raised issues.